# Cypsela and Pappus Morphology and Their Significance for the Taxonomic Delimitation of the Genus *Saussurea* DC. *s*.*str*. and Its Allied Genera (Asteraceae)

**DOI:** 10.3390/plants13233367

**Published:** 2024-11-29

**Authors:** Rubina Abid, Muhammad Munir, Sana Riaz, Muhammad Qaiser

**Affiliations:** 1Department of Botany, University of Karachi, Karachi 75270, Pakistan; rubina@uok.edu.pk (R.A.); sanariazahmed@uok.edu.pk (S.R.); 2Date Palm Research Center of Excellence, King Faisal University, Al-Ahsa 31982, Saudi Arabia; 3Centre for Plant Conservation, University of Karachi, Karachi 75270, Pakistan; qaismd@gmail.com

**Keywords:** Asteraceae, cypsela, pappus, microscopy, *Saussurea*, Pakistan

## Abstract

The cypsela and pappus macro- and micromorphological features of 32 taxa belonging to *Saussurea* DC. *s.str*., *Lipschitziella* Kamelin, *Himalaiella* Raab-Straube, *Dolomiaea* DC., *Aucklandia* Falc., *Frolovia* (DC.) Lipsch. and *Shangwua* Yu J. Wang, Raab-Straube, Susanna, and J.Quan Liu of the tribe Cardueae (Asteraceae) were studied through light and scanning electron microscope to assess the cypsela features of the studied taxa from Pakistan and Kashmir. The cypsela morphological data could also help to discover the taxonomic relationship, as there were no specific and detailed reports available of all the taxa reported from the area under consideration. Various cypsela features, like pappus series, cypsela shape, and surface patterns, were found to be the most significant characters for the taxonomic delimitation of *Saussurea s.l*. The genus *Saussurea s.str*. was delimited from its allied genera due to having biseriate pappus, while the remaining genera pappus were either uniseriate or multiseriate; among these genera, *Dolomiaea* was characterized due to having multiseriate pappus, while the remaining genera, such as *Lipschitziella*, *Himalaiella*, *Aucklandia*, *Shangwua*, and *Frolovia*, had uniseriate pappus. Furthermore, these genera could be delimited based on cypsela shape and surface patterns. Similar to the generic delimitation, cypsela micro and macromorphological characters were also found to be useful for specific delimitation within the studied genera. Most of the cypsela morphological variables when analyzed numerically also proved the taxonomic affiliation for most of the taxa of the genus *Saussurea* and its allied genera. Similarly, these cypsela features could be well correlated with the gross morphological and molecular evidence at the generic and partially for the specific and infraspecific delimitation of *Saussurea s.l.* from Pakistan and Kashmir.

## 1. Introduction

*Saussurea* DC. (Cardueae–Asteraceae), established by de Candolle in 1810 [1] with broad circumscription, is the largest and most heterogenous genus with about 493 species [2], mainly distributed in the temperate regions of Asia, Europe, North America, and Australia [3]. In Pakistan, *Saussurea s.str.* is represented by 29 species [4] spreading over Chitral, Swat, KPK, Baltistan, Gilgit, and Kashmir, while some taxa are also distributed in Baluchistan. In some earlier taxonomic treatments, the concept of the genus was rather diverse, and various segregated taxa were placed under the broad canopy of *Saussurea s.l.* [5,6,7,8,9].

In the present paper, following Lipschitz [3], Ghafoor et al. [4], Raab-Straube [2,10], Susanna et al. [11], and Wang et al. [12], the genus *Saussurea s.str.* has been accepted along with six allied genera, viz., *Frolovia*, *Shangwua*, and *Aucklandia* with single species, while *Lipschitziella* and *Dolomiaea* comprise two species, and *Himalaiella* has seven species, mainly distributed in northern areas of Pakistan.

Cypsela and pappus features have always received due attention for the taxonomic delimitation of various taxa for the family Asteraceae [13,14,15,16,17,18,19,20,21,22,23,24,25,26,27,28]. Saklani et al. [29] studied 23 species of the genus *Saussurea* for their cypsela morphology, but the data were not correlated with the accepted infrageneric classification. Similarly, Ghimire et al. [30] studied the cypsela morphology of 23 species of *Saussurea* from Korea, and data were used for the specific delimitation. Recently, Ghafoor et al. [4] in the Flora of Pakistan and Qaiser et al. [31] utilized the pappus series to distinguish *Saussurea s.str.* from its allied genera. It is evident from the literature, though cypsela morphological studies on *Saussurea* were conducted in different parts of the world, but there are no specific reports available of all the taxa from Pakistan and Kashmir.

Therefore, the present study was undertaken to provide complete information on the macro- and micromorphology of cypsela and pappus of the genus *Saussurea s.str.* and its allied genera with the objective to use these characters as an additional tool for the taxonomic delimitation of *Saussurea s.l.* at the generic as well as specific and infraspecific levels from Pakistan and Kashmir.

## 2. Results

### 2.1. General Cypsela and Pappus Characteristics of the Genus Saussurea s.l.

Cypsela: oblong without or with a depression, curved oblong, oblong–obconic, linear–oblong, oblong–lanceolate, obconical, curved obconical, ovate, broadly ovate, fusiform–oblanceolate, obovate, linear, oblanceolate, with or without horns, sometimes with longitudinal grooves, apical crown toothed, indistinctly or roughly denticulate, denticulate–lacerate, mucronate, sinuate, 0.7–11 × 0.4–3 mm; surface: glabrous or hairy, ribbed or not ribbed, scalariform, lineate, lineolate, ruminate, rugose, appressed colliculate, colliculate in scalariform manner, colliculate in reticulate manner, sulcate, reticulate, falsifoveate, porate, favulariate; color: brown sometimes with dark brown patches or black spots, dark brown–black, brown–purplish, greenish brown, blackish brown, sandy brown, dark grey, and straw-colored; pappus: uniseriate, biseriate, or multiseriate, scabrid, barbellate, plumose, 1.4–30 mm, with a variety of white and brown colors; carpopodium: developed, poorly developed or undeveloped, circular, oblong, pentagonal, ovate, irregular, elliptic, slightly curved, linear, basal or sub-basal, 174–1688 µm, foramen 90–938 µm in diameter.

Key to the genera based on cypsela and pappus features:
1+Pappus biseriate*Saussurea*
Pappus uniseriate or multiseriate22+Pappus multiseriate*Dolomiaea*
Pappus uniseriate33+Cypsela four angled4
Cypsela not four angled54+Cypsela dark grey with four distinct horns on the top*Lipschitziella*
Cypsela brown without horns on the top*Himalaiella*5+Cypsela oblong–oblanceolate, 6–11 mm long*Aucklandia*
Cypsela oblong, curved oblong or obconical, 3.3–4.8 mm long66+Cypsela surface irregularly lineate and favulariate*Shangwua*
Cypsela surface appressedly reticulate and falsifoveate*Frolovia*

### 2.2. General Cypsela and Pappus Characteristics of the Genus Saussurea s.str.

Cypsela: oblong, oblong–obconic, curved oblong, oblong with a depression, obconical, curved obconical, obovate, linear, oblanceolate, 0.7–5 × 0.4–2 mm; surface: glabrous or hairy, ribbed or not ribbed, scalariform, appressedly scalariform, faint scalariform, lineate, lineolate, irregularly lineolate, appressedly lineate, ruminate, rugose, irregularly sulcate, favulariate; color: dark brown, light brown, golden brown, greenish brown, blackish brown, sandy brown or pale brown; pappus: biseriate, 3–17 mm, brownish, pale brown, golden, golden brown, golden white, white, dirty white; carpopodium: developed, poorly developed or undeveloped, oblong, pentagonal, ovate, irregular, elliptic, slightly curved, linear, basal or sub-basal, 174–923 µm, diameter of foramen of carpopodiun 90–515 µm (Table 1, Table 2, Table 3 and Table 4, Figure 1, Figure 2, Figure 3, Figure 4A–F and Figure 5E–J).

The genus is represented here by 17 species, as follows: *Saussurea obvallata* (DC.) Sch.Bip., *S. candolleana* (DC.) Wall. ex Sch.Bip., *S. falconeri* Hook.f., *S. taraxacifolia* (Lindl. ex Royle) Wall. ex DC., *S. devendrae* Pusalkar, *S. andryaloides* (DC.) Sch.Bip., *S. atkinsonii* C.B.Clarke, *S. chondrilloides* C.Winkl., *S. leptophylla* Hemsl., *S. simpsoniana* (Fielding and Gardener) Lipsch., *S. glacialis* Herder, *S. gnaphalodes* (Royle ex DC.) Sch.Bip., *S. bracteata* Decne., *S. schultzii* Hook.f., *S. elliptica* C.B.Clarke ex Hook.f., *S. roylei* (DC.) Sch.Bip., and *S. schlagintweitii* Klatt.

Key to the species of the genus *Saussurea s.str.* based on cypsela and pappus features:
1+Cypsela ribbed2
Cypsela not ribbed72+Cypsela hairy3
Cypsela glabrous43+Cypsela linear, oblanceolate, surface appressed scalariform, or lineolate*S. chondriolloides*
Cypsela oblong, surface rugose*S. schultzii*4+Cypsela golden brown or light brown*S. candolleana*
Cypsela dark brown/brown55+Cypsela curved obconical*S. taraxacifolia*
Cypsela oblong/curved oblong66+Pappus plumose*S. obvallata*
Pappus scabrid*S. roylei*7+Cypsela hairy*S. falconeri*
Cypsela glabrous88+Carpopodium poorly developed or not developed9
Carpopodium well developed109+Cypsela curved oblong, surface ruminate, carpopodium poorly developed*S. andryalloides*
Cypsela obvate, surface rugose, carpopodium absent*S. atkinsonii*10+Carpopodium linear11
Carpopodium elliptic or oblong1211+Cypsela obconical, 5–6 mm long*S. bracteata*
Cypsela oblong with a depression, 3–5 mm long*S. elliptica*12+Carpopodium oblong13
Carpopodium elliptic1413+Cypsela surface appressed reticulate and rugose*S. devendrae*
Cypsela surface irregularly sulcate and favulariate*S. glacialis*14+Cypsela surface rugose and irregularly lineolate*S. simpsoniana*
Cypsela surface not rugose and irregularly lineolate1515+Pappus white*S. leptophylla*
Pappus pale brown/brown1616+Cypsela brown*S. gnaphalodes*
Cypsela pale brown*S. schlagintwitii*

### 2.3. General Cypsela and Pappus Characteristics of Lipschitziella

Cypsela: oblong with four horns, with or without distinct longitudinal grooves; surface: lineate, scalariform, reticulately sulcate; color: dark grey, 2.3–5 × 0.7–0.8 mm, glabrous, not ribbed; pappus: uniseriate, plumose, 11–12 mm, dirty white, carpopodium developed or not developed, circular, sub-basal, ca. 362 µm in diameter, foramen of carpopodium ca. 290 µm in diameter (Table 1, Table 2, Table 3 and Table 4, Figure 4G–J and Figure 5A,B).

The genus *Lipschitziella* is represented here with two species, namely *L. congesta* M. Qaiser, A. Ghafoor, and R. Abid and *L. ceratocarpa* (Decne.) Kamelin. Due to the overlapping of cypsela and pappus characters key to the species could not be constructed.

Further, *L. congesta* has two varieties, viz., *L. congesta* var. *pinnatisecta* R. Abid and M. Qaiser and *L. congesta* var. *congesta*. Similarly, *L. ceratocarpa* has two varieties, i.e., *L. ceratocarpa* var. *ceratocarpa* and *L. ceratocarpa* var. *astorii* R. Abid and M. Qaiser.

Key to the varieties of *Lipschitziella* based on cypsela and pappus features:
1+Cypsela without four longitudinal grooves, surface lineate, scalariform, or lineolate scalariform*L. congesta* var. *pinnetisecta*, *L. ceratocarpa* var. *astorii*, *L. ceratocarpa* var. *ceratocarpa*
Cypsela with four longitudinal grooves, surface reticulately sulcate*L. congesta* var. *congesta*


### 2.4. General Cypsela and Pappus Characteristics of the Genus Himalaiella

Cypsela: four angled, linear–oblong, oblong, oblong–lanceolate, oblong–obconical, ovate, apical crown toothed, indistinctly or roughly denticulate, denticulate–lacerate, mucronate, sinuate; 1.5–6.5 × 0.4–3 mm; surface: rugose, appressed colliculate, colliculate in scalariform manner, colliculate in reticulate manner, sulcate, lineate, irregularly lineate, lineolate, favulariate; color: brown, dark brown, dark brown–black, glabrous, ribbed or not ribbed; pappus: uniseriate, plumose, 1.2–15 mm, white or creamy–white; carpopodium: developed, poorly developed or undeveloped, circular, sub-basal, ca. 321 µm in diameter, foramen 143–333 µm in diameter (Table 1, Table 2, Table 3 and Table 4, Figure 6, Figure 7 and Figure 8G–L).

The genus *Himalaiella* is represented here by six species, viz., *H. chitralica* (Duthie) Raab-Straube, *H. diffusa* (Lipsch.) A. Ghafoor, R. Abid, and M. Qaiser, *H. albescens* (DC.) Raab-Straube, *H. chenopodifolia* (Klatt) Raab-Straube, *H. heteromalla* (D. Don) Raab-Straube, and *H. afghana* (Lipsch.) Raab-Straube.

Key to the species of the genus *Himalaiella* based on cypsela and pappus features:
1+Cypsela ribbed2
Cypsela not ribbed52+Cypsela ovate, surface sulcate, and lineate*H. chitralica*
Cypsela linear–oblong, oblong–obconical, surface lineolate, colliculate in reticulate, or in scalariform manner33+Cypsela with sinuate apical crown, surface lineolate*H. diffusa*
Cypsela with non-sinuate apical crown, surface other than lineolate44+Cypsela with mucronate apical crown, surface colliculate in scalariform manner*H. albescens*
Cypsela with roughly denticulate apical crown, surface colliculate in reticulate manner*H. chenopodifolia*5+Carpopodium circular, well developed*H. heteromalla*
Carpopodium poorly developed*H. afghana*

### 2.5. General Cypsela and Pappus Characteristics of the Genus Dolomiaea

Cypsela: broadly ovate, fusiform–oblanceolate; 3.8–6 × 1–2.9 mm; surface: reticulate and porate; color: brown with dark brown patches, hairy or glabrous, ribbed or not ribbed, pappus multiseriate, barbellate, 15–30 mm, brown–purplish, creamy–white; carpopodium: irregular, basal, ca. 1688 µm in diameter, foramen of carpopodium ca. 938 µm in diameter (Table 1, Table 2, Table 3 and Table 4, Figure 4L, Figure 8E,F and Figure 9A–D).

The genus *Dolomaiaea* is represented here by two species, viz., *D. macrocephala* Royle and *D. megacephala* M. Qaiser, A. Ghafoor, and Roohi B. Abid.

Key to the species of the genus *Dolomaiaea* based on cypsela and pappus features:
1+Cypsela ribbed, fusiform–oblanceolate*D. macrocephala*
Cypsela not ribbed, broadly ovate*D. megacephala*

### 2.6. General Cypsela Characteristics of the Genus Shangwua

Cypsela: curved oblong, 3.9–4.8 × 0.8–1 mm, irregularly lineate and favulariate, glabrous, ribbed, light brown; pappus: uniseriate, plumose, 1.4–1.5 mm, dirty white–brownish; carpopodium circular, basal, ca. 464 µm in diameter, foramen of carpopodium ca. 357 µm in diameter (Table 1, Table 2, Table 3 and Table 4, Figure 5C,D and Figure 9E–G).

The genus *Shangwua* is represented here by a single species, viz., *S. jacea* (Klotzsch) Yu J. Wang and Rabb-Straube.

### 2.7. General Cypsela and Pappus Characteristics of the Genus Frolovia

Cypsela: oblong, 3.3–3.9 × 0.8–1.2 mm, appressedly reticulate and falsifoveate, straw colored, glabrous, ribbed; pappus: uniseriate, plumose, 14–15 mm; color: pale white; carpopodium: oblong, basal, ca. 500 µm in diameter, foramen of carpopodium ca. 341 µm in diameter (Table 1, Table 2, Table 3 and Table 4, Figure 8C,D and Figure 9H–J).

The genus *Frolovia* is represented here by a single species, viz., *F. gilesii* (Hemsl.) B.A. Scheripova.

### 2.8. General Cypsela and Pappus Characteristics of the Genus Aucklandia

Cypsela: oblong–lanceolate, 6–11 × 1.1–1.5 mm lineate, sulcate and faintly scalariform, a stomata was found on surface, brown with black spots, glabrous, ribbed; pappus: uniseriate, plumose, 1.2–1.5 mm; color: brownish; carpopodium: circular, basal, 600 µm in diameter, foramen of carpopodium c. 529 µm in diameter (Table 1, Table 2, Table 3 and Table 4, Figure 8A,B and Figure 9K,L).

The genus *Aucklandia* is represented here by a single species, viz., *A. costus* Falc.

## 3. Discussion

Cypsela morphology has long been used by different workers for the taxonomic delimitation of various taxa within the family Asteraceae [22,23,25,26,27,28,30,32,33,34,35,36]. In the present study, the cypsela morphology of 32 taxa belonging to *Saussurea s.str.* and its six allied genera (Cardueae–Asteraceae) from Pakistan and Kashmir were studied (Table 1, Table 2 and Table 3, Figure 1, Figure 2, Figure 3, Figure 4, Figure 6, Figure 7 and Figure 9). Similar to the previous findings on Asteraceous taxa [24,26,37,38], it was found that pappus series, cypsela shape, and surface patterns were the most variable and significant features for the taxonomic delimitation of various taxa of *Saussurea s.l.*, such as the genus *Saussurea s.str*, which remained distinct from all the allied genera by its biseriate pappus, while the pappus of the remaining genera were either uniseriate or multiseriate. Among them, *Dolomiaea* was distinguished by multiseriate pappus, while the remaining genera, i.e., *Lipschitziella*, *Aucklandia*, *Shangwua*, *Himalaiella*, and *Frolovia*, had uniseriate pappus. Furthermore, the genera *Lipschitziella* and *Himalaiella* were separated from the other three genera in the uniseriate group by having quadrangular cypsela. They remained distinct from each other by dark grey cypsela with four horns in former and brown cypsela without horns in latter, while the remaining three genera can be distinguished by their cypsela shapes, size, and surface patterns, such as *Aucklandia*, had oblong–oblanceolate and larger (6–11 mm) cypselas. The remaining two genera, i.e., *Shangwua* and *Frolovia*, had smaller (3.3–4.8 mm) cypselas, but both the genera had distinct cypsela surfaces, i.e., irregularly lineate and favulariate surface in the former and appressed reticulate and falsifoveate surface in the latter one. Our findings were also in accordance with the results of phylogenetic studies of Raab-Straube [10] who established a new genus *Himalaiella* and accepted *Frolovia* and *Lipschitziella* as distinct genera. This was followed by Shi and Raab-Straube [39] and Ghafoor et al. [4], in Flora of China and Flora of Pakistan, respectively, and also accepted *Himalaiella* and *Frolovia* as separate genera mainly based on pappus series. The cypsela morphological features fully support the generic delimitation of *Saussurea s.l.* based on gross morphology [4] and molecular studies [40].

Similarly, cluster analysis also proved the delimitation of *Saussurea* from its allied genera. Two main groups were formed in dendrogram one, having all the species of *Saussurea*, while the remaining genera fell in another group (Figure 10 and Figure 11).

Group I was further divided into two subgroups, i.e., I_a_ and I_b_. Subgroup I_a_ comprised four varieties belonging to two species of *Lipschitziella*. They grouped together by having dark grey cypsela with four horns at the top. While group I_b_ was further split into three more subgroups. One of them comprised two species of *Dolomiaea*, which are grouped together due to their multiseriate pappus but are distinguished by a ribbed cypsela in *D. macrocephala* and a ribless cypsela in *D. megacephala*. In the second subgroup of group I_b_, *Frolovia gilesii* remains distinct from *Shangwua jacea*, *Aucklandia costus*, and *Himalaiella heteromalla* due to its oblong carpopodium. While they remain distinct from each other due to their distinct cypsela color, shape, and surfaces. The third group in group I_b_ comprised of all the remaining species of *Himalaiella*. Among them, *H. chitralica* and *H. chenopodifolia* were closed to each other due to the denticulate apical crown and distinguished due to their distinct surfaces; further, *H. afghana*, *H. diffusa*, and *H. albescens* were grouped together due to their linear–oblong cypsela but remained separated by their nonribbed cypsela with a denticulate–lacerate apical crown in *H. afghana*. While ribbed cypsela with sinuate and mucronate apical crowns in *H. diffusa* and *H. albescens*, respectively, and distinct cypsela surfaces. Similar to our findings, Susanna et al. [11], Kasana et al. [40], and Kita et al. [41] also pointed out close affinities of *Himalaiella* with *Lipschitziella* based on morphological and molecular data.

On the basis of cypsela surface, group II was also bifurcated into two subgroups i.e., subgroup II_a_ and subgroup II_b._ Group II_a_ was further split into two clads. One of them comprised *Saussurea taraxacifolia* and *S. elliptica* due to the golden–white pappus and remained separated by the sub-basal carpopodium in the former and the basal carpopodium in the latter. The other group had a golden or pale brown pappus, comprising *Saussurea gnaphalodes*, *S. schlegintweitii*, *S. obvallata*, and *S. roylei*. Among them, *S. gnaphalodes* and *S. schlegintweitii* were grouped together due to their similar cypsela surface, viz., appressedly lineate and scalariform. On the other hand, the remaining taxa were separated by the oblong–obconic cypsela shape in *S. gnaphalodes* and oblong cypsela in *S. schlegintweitii*. Similarly, *S. obvallata* and *S. roylei* were grouped together due to ribbed cypsela and remained distinct by a sub-basal, oblong carpopodium in the former and a circular carpopodium in the latter. Our findings are also in accordance with Raab-Straube [10] who used the carpopodium shape as the diagnostic character within the genus *Saussurea*. In group II_b_, *S. andryaloides* and *S. chodrioloides* fell apart from all other species due to their poorly developed carpopodium. On the other hand, they were separated from each other by their specific cypsela shape and surface. The remaining species further formed two clads. One comprised of three species, viz., *S. glacialis*, *S. bracteate*, and *S. candolleana*, but among them, *S. candolleana* was separate due to its lineate surface and pentagonal carpopodium. Moreover, *S. glacialis* and *S. bracteata* were different due to the presence of an oblong cypsela and carpopodium and an obconical cypsela with linear carpopodium, respectively. The rest of the six species, namely *S. simpsoniana*, *S. schultzii*, *S. atkinsonii*, *S. devendrae*, *S. falconeri*, and *S. leptophylla* fell in the last clad; amongst them, *S. simpsoniana* and *S. schultzii* were grouped together due to the golden brown cypsela, but still, both the species remain distinct by having an elliptic carpopodium with basal hilum and irregular carpopodium with a sub-basal hilum, respectively. Further, the alliance of *S. atkinsonii*, *S. devendrae*, *S. falconeri*, and *S. leptophylla* may be due to having a nonribbed cypsela and plumose pappus bristles. While *S. atkinsonii* remains distinct from rest of the species due to the obovate cypsela and the absence of carpopodium. Similarly, the cypsela of *S. devendrae* exclusively had an appressedly reticulate and rugose surface pattern and golden brown pappus bristles. The remaining last two species, *S. falconeri* and *S. devendrae*, remain distinct by having hairy and greenish brown cypsela in the former and glabrous along with pale brown cypsela in the latter. From the aforesaid discussion, it is noteworthy that some taxa of *Saussurea* and *Lipschitziella* showed an exclusive alliance rather than gross morphology (Figure 12). However, a previous study found that numerical analysis does not always fully support the gross morphological data, which our findings also support [42].

Thus, from the above discussion, it can be concluded that cypsela morphological features, like shape, carpopodium, and sculpturing patterns, almost completely support the gross morphological and molecular decisions made for the taxonomic delimitation of the genus *Saussurea* and its allied genera. However, most of the species belonging to *Saussurea* and its allied genera can easily be delimited on the basis of cypsela morphology.

## 4. Materials and Methods

Thirty-two taxa belonging to 7 genera, i.e., *Saussurea*, *Lipschitziella*, *Himalaiella*, *Dolomiaea*, *Aucklandia*, *Frolovia*, and *Shangwua* from Pakistan and Kashmir were studied for cypsela and pappus characteristics. A total of thirty mature and healthy cypselas per specimen, with ten specimens per species, were included in this study (Appendix B). The macro- and micromorphological characters of cypsela, i.e., shape, surface, color, size, ribs, pappus (series, structure, length, color), carpopodium shape, position, diameter, and diameter of foramen of carpopodium were examined under a stereo microscope (Nikon XN model, Tokyo, Japan), compound microscope (Nikon Type 102, Tokyo, Japan), and scanning electron microscope (JEOL JSM 5910, JEOL Ltd., Tokyo, Japan). For scanning electron microscopy, mature cypselas were placed on stubs and coated with gold–palladium particles and observed under a scanning electron microscope. Additionally, hierarchical classification was conducted using numerical analysis and the Euclidean distance index, with the help of a statistical software package [43]. Each taxon was considered an operational taxonomic unit (OTU). Qualitative characters were recorded in the binary state as 1, 2, and characters, which were either absent or present and were coded as 0 or 1, respectively. While, for quantitative characters, average values were directly used (Table 4 and Appendix A).

## 5. Conclusions

From the above discussion, it can be concluded that cypsela and pappus morphological features, like shape, carpopodium, and sculpturing patterns, almost completely support the gross morphological and molecular decisions made for the taxonomic delimitation of the genus *Saussurea* and its allied genera. However, most of the species belonging to *Saussurea* and its allied genera can easily be delimited on the basis of cypsela morphology.

## Figures and Tables

**Figure 1 plants-13-03367-f001:**
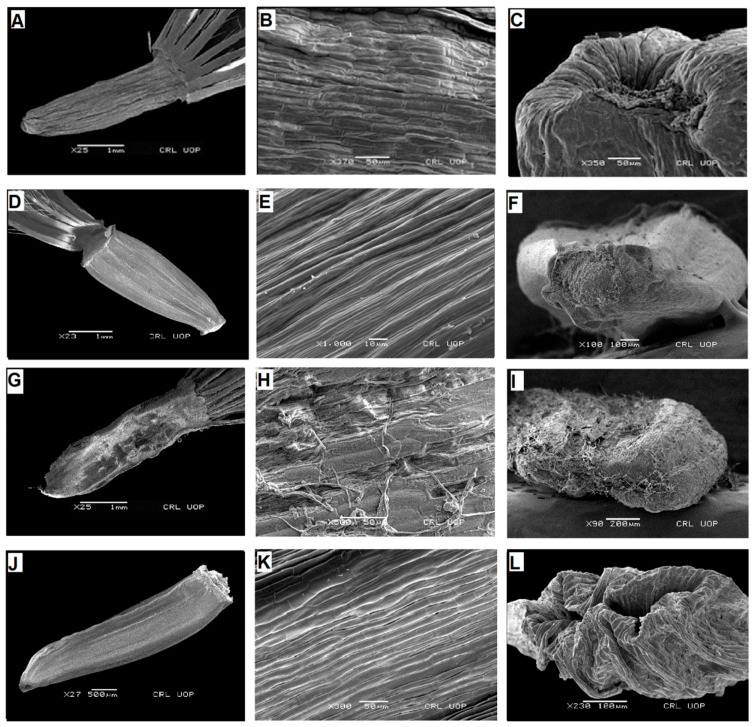
Scanning electron micrographs of cypsela: *Saussurea obvalata* (**A**), cypsela (**B**), cypsela surface (**C**), carpopodium; *S. candolleana* (**D**), cypsela (**E**), cypsela surface (**F**), carpopodium; *S. falconeri* (**G**), cypsela (**H**), cypsela surface (**I**), carpopodium; *S. taraxacifolia* (**J**), cypsela (**K**), cypsela surface (**L**), carpopodium. (Scale bars: (**A**,**D**,**G**) = 1 mm; (**E**) = 10 µm; (**B**,**C**,**H**,**K**) = 50 µm; (**F**,**L**) = 100 µm; (**I**) = 200 µm; (**J**) = 500 µm.).

**Figure 2 plants-13-03367-f002:**
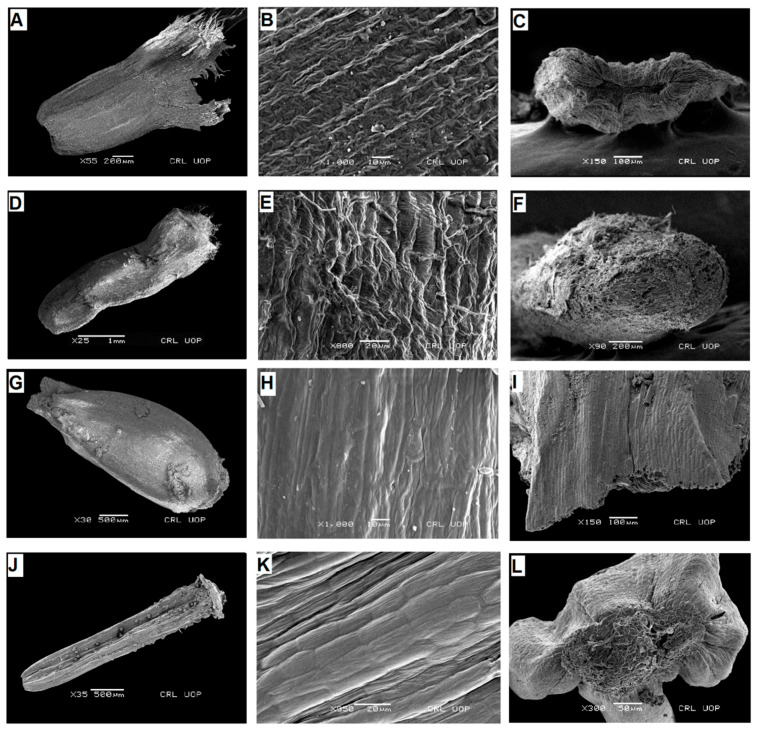
Scanning electron micrographs of cypsela: *Saussurea devendrae* (**A**), cypsela (**B**), cypsela surface (**C**), carpopodium; *S. andryaloides* (**D**), cypsela (**E**), cypsela surface (**F**), carpopodium; *S. atkinsonii* (**G**), cypsela (**H**), cypsela surface (**I**), carpopodium; *S. chondrilloides* (**J**), cypsela (**K**), cypsela surface (**L**), carpopodium. (Scale bars: (**D**) = 1 mm; (**B**,**H**) = 10 µm; (**E**,**K**) = 20 µm; (**L**) = 50 µm; (**C**,**I**) = 100 µm; (**A**,**F**) = 200 µm; (**G**,**J**) = 500 µm.).

**Figure 3 plants-13-03367-f003:**
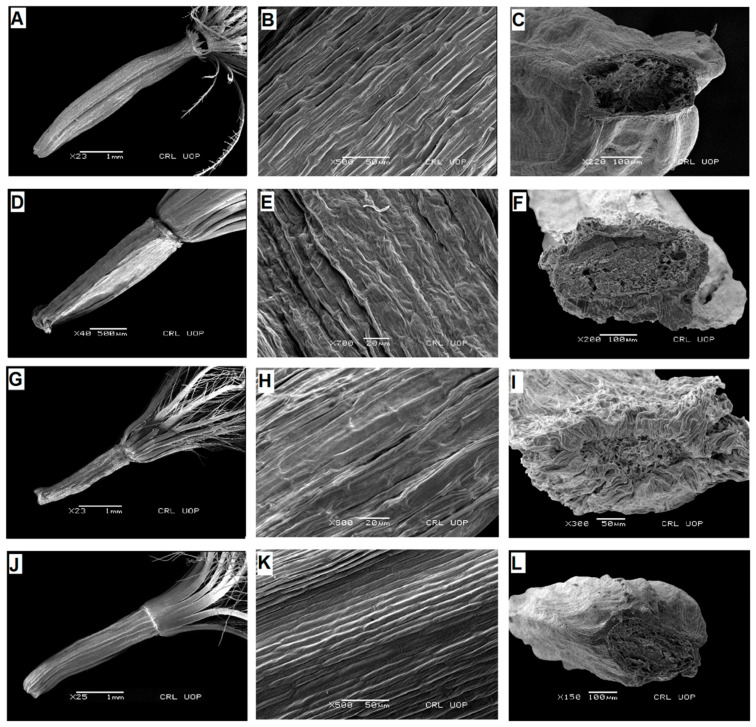
Scanning electron micrographs of cypsela: *Saussurea leptophylla* (**A**), cypsela (**B**), cypsela surface (**C**), carpopodium; *S. simpsoniana* (**D**), cypsela (**E**), cypsela surface (**F**), carpopodium; *S. glacialis* (**G**), cypsela (**H**), cypsela surface (**I**), carpopodium; *S. gnaphalodes* (**J**), cypsela (**K**), cypsela surface (**L**), carpopodium. (Scale bars: (**A**,**G**,**J**) = 1 mm; (**E**,**H**) = 20 µm; (**B**,**I**,**K**) = 50 µm; (**C**,**F**,**L**) = 100 µm; (**D**) = 500 µm.).

**Figure 4 plants-13-03367-f004:**
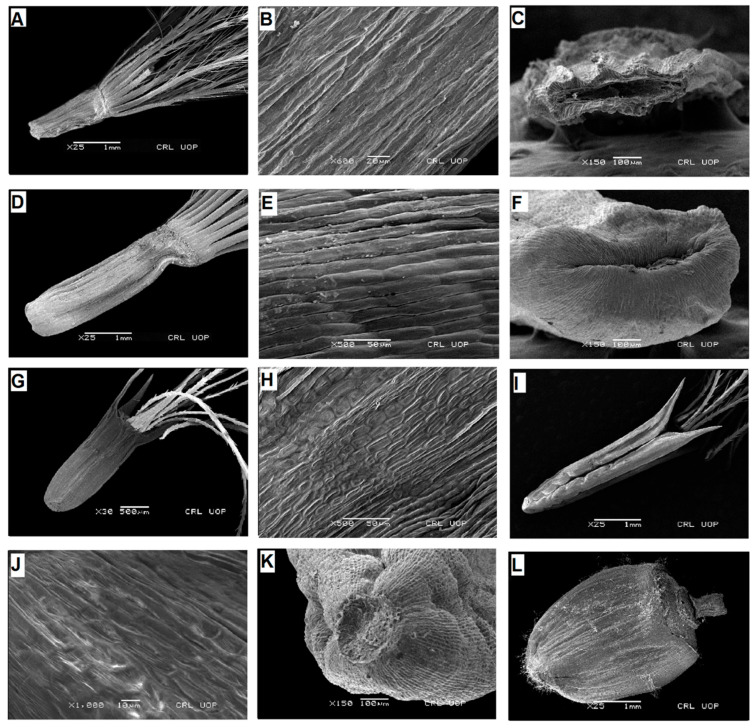
Scanning electron micrographs of cypsela: *Saussurea bracteata* (**A**), cypsela (**B**), cypsela surface (**C**), carpopodium; *S. elliptica* (**D**), cypsela (**E**), cypsela surface (**F**), carpopodium; *Lipschitziella congesta* var. *pinnatisecta* (**G**), cypsela (**H**), cypsela surface; *Lipschitziella congesta* var. *congesta* (**I**), cypsela (**J**), cypsela surface (**K**), carpopodium; *D. megacephala* (**L**), cypsela. (Scale bars: (**A**,**D**,**I**,**L**) = 1 mm; (**J**) = 10 µm; (**B**) = 20 µm; (**E**,**H**) = 50 µm; (**C**,**F**,**K**) = 100 µm; (**G**) = 500 µm.).

**Figure 5 plants-13-03367-f005:**
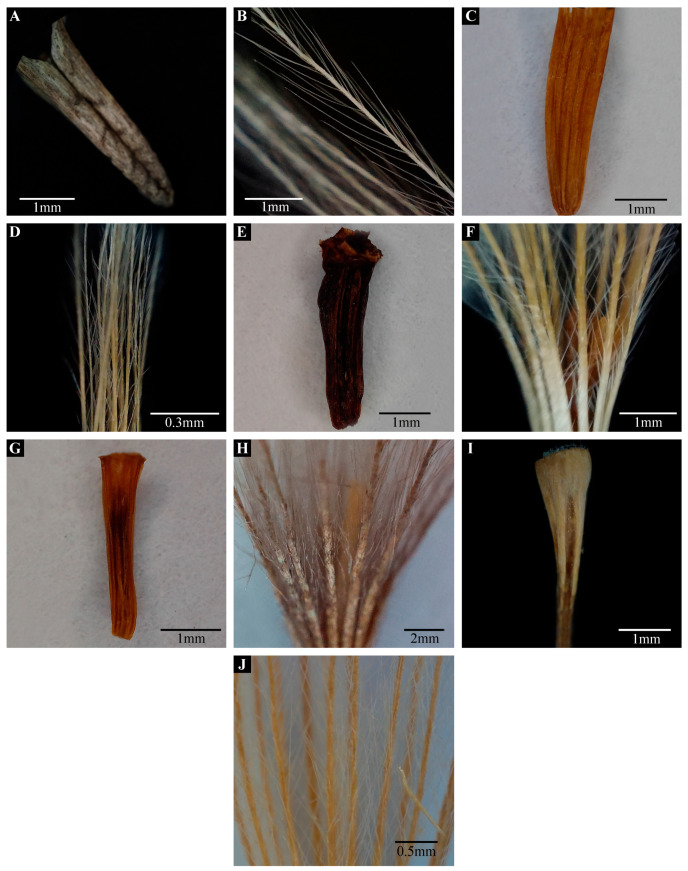
Light micrographs of cypsela and pappus: *Lipschitziella congesta* var. *congesta* (**A**), cypsela (**B**), pappus; *Shangwua jacea* (**C**), cypsela (**D**), pappus; *Saussurea obvallata* (**E**), cypsela (**F**), pappus; *S. gnaphalodes* (**G**), cypsela (**H**), pappus; *S. falconeri* (**I**), cypsela; *S. taraxacifolia* (**J**), pappus. (Scale bar: (**A**–**C**,**E**–**G**,**I**) = 1 mm; (**D**) = 0.3 mm; (**H**) = 2 mm; (**J**) = 0.5 mm).

**Figure 6 plants-13-03367-f006:**
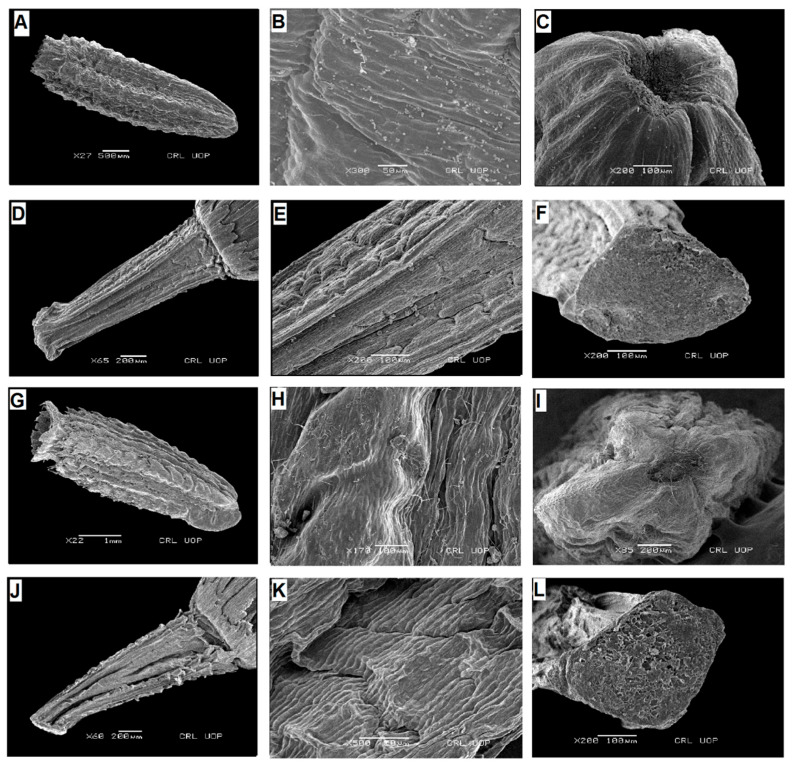
Scanning electron micrographs of cypsela: *Hemalaiella heteromalla* (**A**), cypsela (**B**), cypsela surface (**C**), carpopodium; *H. chitralica* (**D**), cypsela (**E**), cypsela surface (**F**), carpopodium; *H. afghana* (**G**), cypsela (**H**), cypsela surface (**I**), carpopodium; *H. albescens* (**J**), cypsela (**K**), cypsela surface (**L**), carpopodium. (Scale bars: (**G**) = 1 mm; (**B**,**K**) = 50 µm; (**C**,**E**,**F**,**H**,**L**) = 100 µm; (**D**,**I**,**J**) = 200 µm; (**A**) = 500 µm.).

**Figure 7 plants-13-03367-f007:**
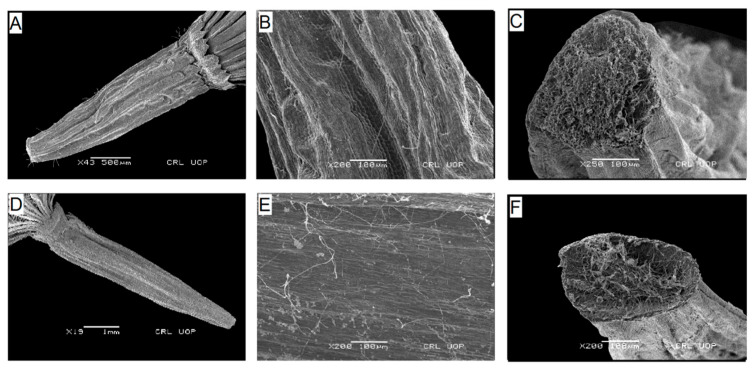
Scanning electron micrographs of cypsela: *Himalaiella chenopodifolia* (**A**), cypsela (**B**), cypsela surface (**C**), carpopodium; *H. diffusa* (**D**), cypsela (**E**), cypsela surface (**F**), carpopodium. (Scale bars: (**D**) = 1 mm; (**B**,**C**,**E**,**F**) = 100 µm; (**A**) = 500 µm.).

**Figure 8 plants-13-03367-f008:**
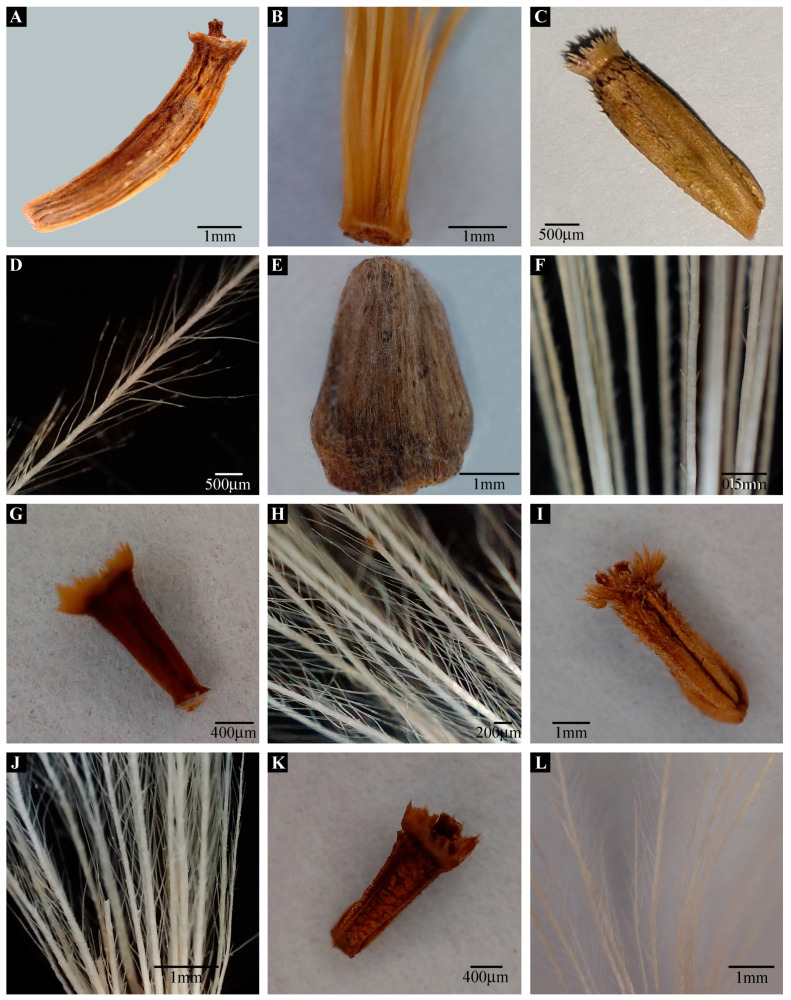
Light micrographs of cypsela and pappus: *Aucklandia costus* (**A**), cypsela (**B**), pappus; *Frolovia gilesii* (**C**), cypsela (**D**), pappus; *Dolomiaea megacephala* (**E**), cypsela (**F**), pappus; *Himalaiella chitralica* (**G**), cypsela (**H**), pappus; *H. afghana* (**I**), cypsela (**J**), pappus; *H. albescens* (**K**), cypsela (**L**), pappus. (Scale bar: (**A**,**B**,**E**,**I**,**J**,**L**) = 1 mm; (**C**,**D**) = 500 µm; (**F**) = 0.5 mm; (**G**,**K**) = 400 µm; (**H**) = 200 µm.).

**Figure 9 plants-13-03367-f009:**
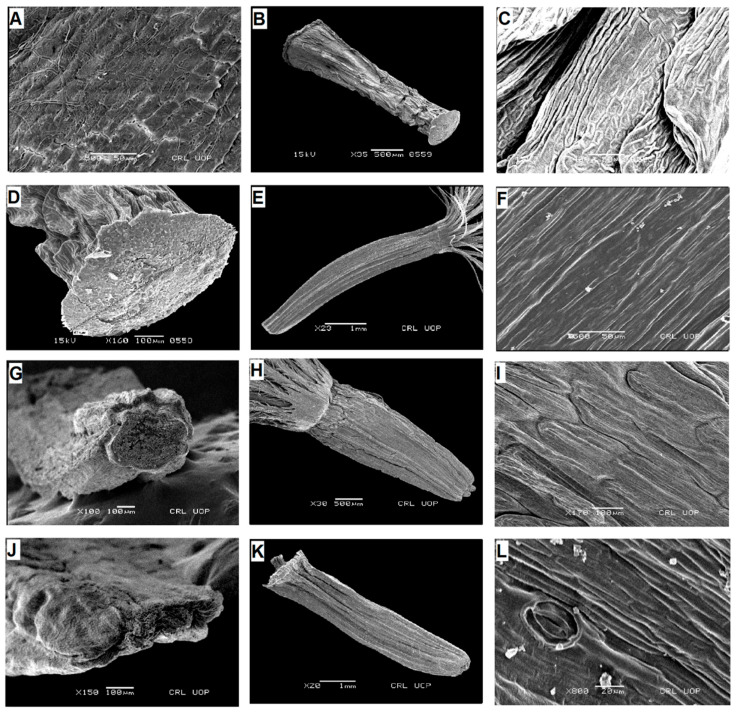
Scanning electron micrographs of cypsela: *Dolomiaea megacephala* (**A**), cypsela surface; *D. macrocephala* (**B**), cypsela (**C**), cypsela surface (**D**), carpopodium; *Shangwua jacea* (**E**), cypsela (**F**), cypsela surface (**G**), carpopodium; *Frolovia gilesii* (**H**), cypsela (**I**), cypsela surface (**J**), carpopodium; *Aucklandia costus* (**K**), cypsela (**L**), cypsela surface. (Scale bars: (**E**,**K**) = 1 mm; (**L**) = 20 µm; (**A**,**C**,**F**) = 50 µm; (**D**,**G**,**I**,**J**) = 100 µm; (**B**,**H**) = 500 µm.).

**Figure 10 plants-13-03367-f010:**
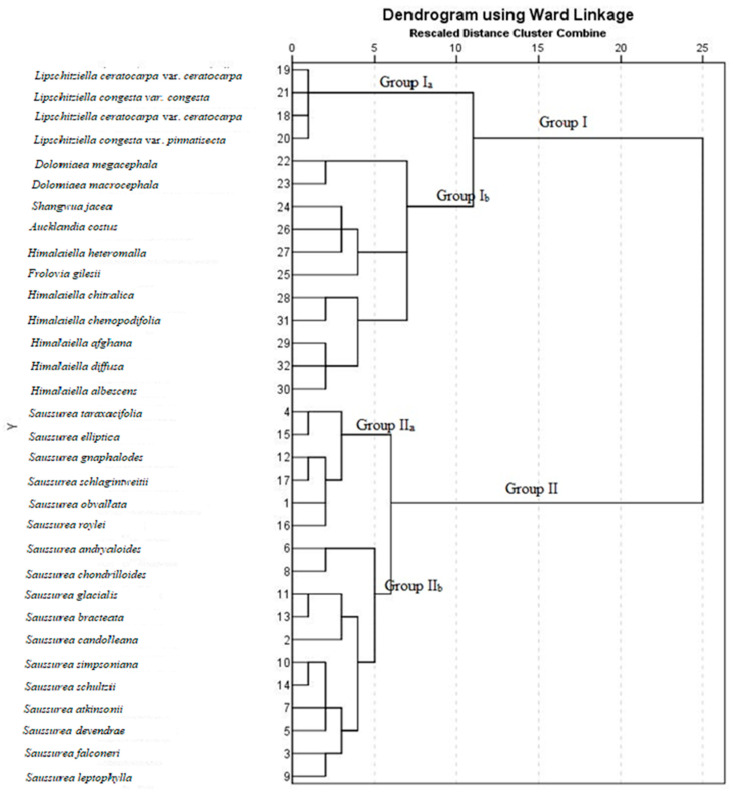
Dendrogram showing the relationship of the taxa of the genus *Saussurea* and its allied genera.

**Figure 11 plants-13-03367-f011:**
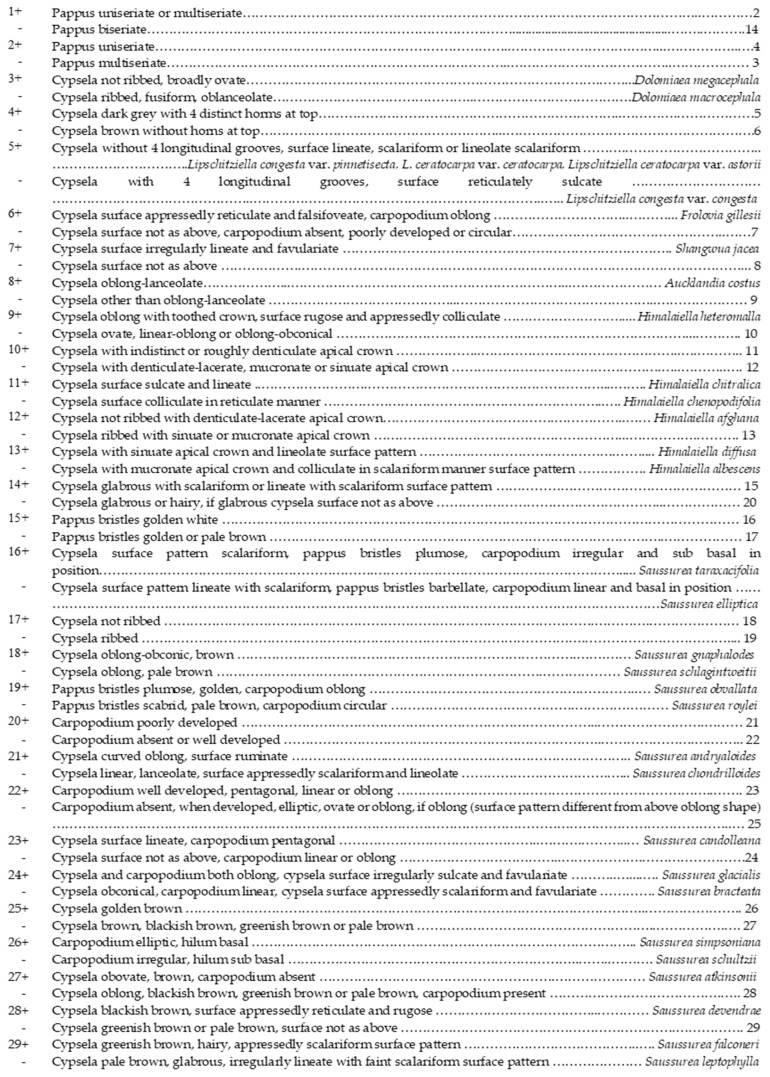
Synthetic key for the taxa of *Saussurea* (s. str.) and its allied genera based on cypsela and pappus features.

**Figure 12 plants-13-03367-f012:**
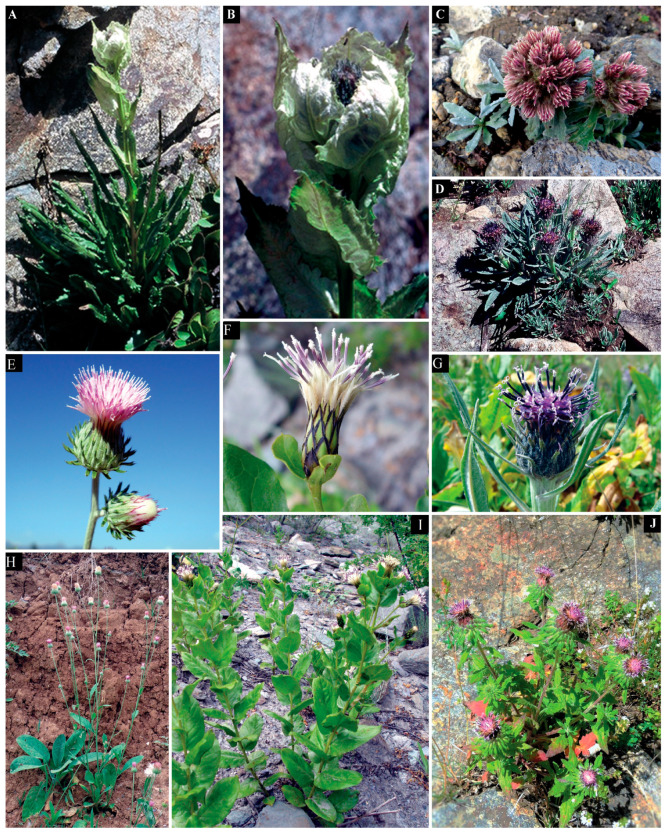
*Saussurea obvallata*: (**A**) habit; (**B**) synflorescence laxly enveloped by connivent leaves. *Saussurea gnaphalodes*: (**C**) habit; *Saussurea falconeri*: (**D**) habit; (**G**) capitulum. *Himalaiella heteromalla*: (**H**) habit; (**E**) capitulum. *Shangwua jacea*: (**I**) habit; (**F**) capitulum. *Lipschitziella ceratocarpa* var. *ceratocarpa*: (**J**) habit. (Source: Flora of Pakistan, Asteraceae (V)-Cardueae: 223. Centre of Plant Conservation, University of Karachi, Karachi, Pakistan).

**Table 1 plants-13-03367-t001:** Cypsela characteristics of the genus *Saussurea* and its allied genera.

S. No.	Name of Taxa	Cypsela						
		Shape	Surface	Color	Length (mm)	Breadth (mm)	Hairy/Glabrous	Ribs
1	*Saussurea obvallata*	Oblong	Scalariform	Dark brown	3.2 (3.5) 3.8 ± 0.21	0.9 (1) 1.1 ± 0.07	Glabrous	Ribbed
2	*S. candolleana*	Obconical	Lineate	Light brown or golden brown	1.6 (2.8) 4 ± 0.76	0.8 (1) 1.2 ± 0.14	Glabrous	Ribbed
3	*S. falconeri*	Oblong	Appressedly Scalariform	Greenish brown	3.8 (3.9) 4.1 ± 0.10	1.2 (1.24) 1.3 ± 0.05	Hairy	Not ribbed
4	*S. taraxacifolia*	Curved obconical	Appressedly lineate and scalariform	Dark brown	3.5 (4.2) 4.8 ± 0.50	0.9 (1.1) 1.3 ± 0.13	Glabrous	Faintly ribbed
5	*S. devendrae*	Oblong	Appressedly reticulate and rugose	Blackish brown	1.3 (1.5) 1.6 ± 0.10	0.5	Glabrous	Not ribbed
6	*S. andryaloides*	Curved oblong	Ruminate	Sandy brown	3.9 (3.92) 4 ± 0.04	1.0 (1.2) 1.50.16	Glabrous	Not ribbed
7	*S. atkinsonii*	Obovate	Rugose	Brown	3.5 (3.9) 4.2 ± 0.23	1.9 (2) 2.1 ± 0.08	Glabrous	Not ribbed
8	*S. chondrilloides*	Linear, oblanceolate	Appressedly scalariform and lineolate	Brown	3.1 (3.3) 3.5 ± 0.18	0.6 (0.7) 0.74 ± 0.04	Sparsely hairy	Ribbed
9	*S. leptophylla*	Oblong	Irregularly lineate with faint scalariform pattern	Pale brown	4 (4.5) 5 ± 0.41	0.7 (0.78) 0.8 ± 0.04	Glabrous	Not ribbed
10	*S. simpsoniana*	Oblong	Rugose and irregularly lineolate	Golden brown	2.4 (2.5) 2.8 ± 0.12	0.8	Glabrous	Not ribbed
11	*S. glacialis*	Oblong	Irregularly sulcate and favulariate	Brown	2.1 (2.2) 2.3 ± 0.08	0.4 (0.48) 0.5 ± 0.04	Glabrous	Not ribbed
12	*S. gnaphalodes*	Oblong–obconic	Appressedly lineate and Scalariform	Brown	3 (3.5) 4 ± 0.52	0.5	Glabrous	Not ribbed
13	*S. bracteata*	Obconical	Appressedly scalariform and favulariate	Brown	1.1 (1.4) 1.75 ± 0.25	0.6 (0.7) 0.8 ± 0.08	Glabrous	Not ribbed
14	*S. schultzii*	Oblong	Rugose	Golden brown	0.7 (0.8) 1 ± 0.15	0.2 (0.25) 0.3 ± 0.04	Hairy	Ribbed
15	*S. elliptica*	Oblong with a depression	Appressedly lineate and scalariform	Blackish brown	3.7 (3.79) 3.8 ± 0.03	1	Glabrous	Not ribbed
16	*S. roylei*	Curved oblong	Lineate and scalariform	Brown	4 (5) 7 ± 1.05	1.1 (1.2) 1.3 ± 0.08	Glabrous	Ribbed
17	*S. schlagintweitii*	Oblong	Appressedly lineate and scalariform	Pale brown	4 (4.5) 4.8± 0.35	1.1 (1.2) 1.5 ± 0.14	Glabrous	Not ribbed
18	*Lipschitziella ceratocarpa* var. *ceratocarpa*	Quadrangular with 4 horns	Appressedly scalariform and lineolate	Dark grey	5 (5.5) 5.8 ± 0.29	1 (1.5) 1.6 ± 0.18	Glabrous	Not ribbed
19	*L. ceratocarpa* var. *astorii*	Quadrangular with 4 horns	Scalariform	Dark grey	7 (7.5) 8 ± 0.52	1.5 (1.8) 2 ± 0.25	Glabrous	Not ribbed
20	*L. congesta* var. *pinnatisecta*	Oblong, quadrangular with 4 horns (up to 1 mm)	Lineate, scalariform	Dark grey	2.3 (3.6) 5 ± 1.01	0.7 (0.8) 0.9 ± 0.09	Glabrous	Not ribbed
21	*L. congesta* var. *congesta*	Oblong, quadrangular with 4 distinct longitudinal grooves and 4 horns (exceeding 1 mm)	Reticulately sulcate	Dark grey	3.5 (3.6) 4 ± 0.18	0.5 (0.7) 0.8 ± 0.12	Glabrous	Not ribbed
22	*Dolomiaea megacephala*	Broadly ovate	Reticulate and porate	Brown with dark brown patches	3.8 (3.9) 4 ± 0.09	2.4 (2.7) 2.9 ± 0.20	Hairy	Not ribbed
23	*D. macrocephala*	Fusiform–oblanceolate	Reticulate and porate	Ash brown with maculated spots	5 (5.5) 6 ± 0.52	1 (1.25) 1.5 ± 0.26	Glabrous	Ribbed
24	*Shangwua jacea*	Curved oblong	Irregularly lineate and favulariate	Light brown	3.9 (4.4) 4.8 ± 0.31	0.8 (0.9) 1 ± 0.09	Glabrous	Ribbed
25	*Frolovia gilesii*	Oblong, obconical	Appressedly reticulate and falsifoveate	Straw colored	3.3 (3.6) 3.9 ± 0.18	0.8 (1) 1.2 ± 0.10	Glabrous	Ribbed
26	*Aucklandia costus*	Oblong–lanceolate	Lineate, sulcate, and faintly scalariform; a stomata was found on surface	Brown with black spots	6 (8.5) 11 ± 1.41	1.1 (1.3) 1.5 ± 0.18	Glabrous	Ribbed
27	*Himalaiella heteromalla*	Oblong with toothed crown, obtusely 4 angled	Rugose and appressedly colliculate	Dark brown–black	3 (3.6) 4.1 ± 0.37	1.1 (1.2) 1.3 ± 0.21	Glabrous	Not ribbed
28	*H. chitralica*	Ovate with apical crown indistinctly denticulate, 4 angled	Sulcate and lineate	Brown	1.5 (1.8) 3 ± 0.27	0.4 (0.45) 0.5 ± 0.04	Glabrous	Ribbed
29	*H. afghana*	Linear–oblong, slightly curved, with apical crown denticulate–lacerate, 4 angled	Irregularly lineate, favulariate, and appressedly colliculate	Brown	4.5 (5.5) 6.5 ± 0.59	1.5 (1.7) 1.9 ± 0.15	Glabrous	Not ribbed
30	*H. albescens*	Linear–oblong, with apical crown mucronate, 4 angled	Colliculate in scalariform manner	Brown	1.7 (2.8) 4 ± 0.89	0.5	Glabrous	Ribbed
31	*H. chenopodifolia*	Oblong–obconical, with apical crown roughly denticulate, roughly tetragonal	Colliculate in reticulate manner	Dark brown	5.5 (6) 6.5 ± 0.28	0.5 (0.8) 1 ± 0.18	Glabrous	Ribbed
32	*H. diffusa*	Linear–oblong, with apical crown sinuate, obtusely 4 angled	Lineolate	Brown	2.5 (2.7) 3 ± 0.17	0.8 (1) 1.2 ± 0.09	Glabrous	Ribbed

The data in the length and breadth columns indicated minimum value (mean value), maximum value, and ± standard deviation. All measurements are in millimeters (mm).

**Table 2 plants-13-03367-t002:** Pappus characteristics of the genus *Saussurea* and its allied genera.

S. No.	Name of Taxa	Pappus			
		Series	Structure	Length (mm)	Color
1	*Saussurea obvallata*	Biseriate	Plumose	3 (9) 15 ± 2.89	Golden
2	*S. candolleana*	Biseriate	Plumose	8 (9) 10 ± 0.09	Golden brown
3	*S. falconeri*	Biseriate	Plumose	12 (14) 17 ± 0.67	Brownish
4	*S. taraxacifolia*	Biseriate	Plumose	13 (14) 16 ± 0.86	Golden white
5	*S. devendrae*	Biseriate	Plumose	6 (7) 8 ± 0.05	Golden brown
6	*S. andryaloides*	Biseriate	Plumose	4 (9) 15 ± 1.59	Golden white
7	*S. atkinsonii*	Biseriate	Plumose	6.5 (10) 13 ± 1.08	Golden white
8	*S. chondrilloides*	Biseriate	Plumose	4 (7) 10 ± 2.45	Golden white
9	*S. leptophylla*	Biseriate	Plumose	3 (7) 12 ± 3.23	White
10	*S. simpsoniana*	Biseriate	Smooth	12 (13) 14 ± 0.54	Golden white
11	*S. glacialis*	Biseriate	Plumose	8 (9) 10 ± 0.05	Golden brown
12	*S. gnaphalodes*	Biseriate	Plumose	9 (9.5) 10 ± 0.05	Pale brown
13	*S. bracteata*	Biseriate	Plumose, barbellate	2 (8) 15 ± 3.48	Dirty white
14	*S. schultzii*	Biseriate	Barbellate	3 (4) 5 ± 0.05	Dirty white
15	*S. elliptica*	Biseriate	Barbellate	10 (13) 15 ± 1.06	Golden white
16	*S. roylei*	Biseriate	Scabrid	8 (12) 15 ± 0.95	Pale brown
17	*S. schlagintweitii*	Biseriate	Scabrid, plumose	12 (13) 14 ± 0.16	Pale brown
18	*Lipschitziella ceratocarpa* var. *ceratocarpa*	Uniseriate	Plumose	8 (10) 12 ± 0.18	Dirty white
19	*L. ceratocarpa* var. *astorii*	Uniseriate	Plumose	10 (11) 12 ± 0.04	Dirty white
20	*L. congesta* var. *pinnatisecta*	Uniseriate	Plumose	11 (11.5) 12 ± 0.45	Dirty white
21	*L. congesta* var. *congesta*	Uniseriate	Plumose	11 (11.5) 12 ± 0.16	Dirty white
22	*Dolomiaea megacephala*	Multiseriate	Barbellate	28 (29) 30 ± 0.09	Brown–purplish
23	*D. macrocephala*	Multiseriate	Barbellate	15 (16) 17 ± 0.05	Creamy–white
24	*Shangwua jacea*	Uniseriate	Plumose	1.4 (1.45) 1.5 ± 0.11	Dirty white– brownish
25	*Frolovia gilesii*	Uniseriate	Plumose	14 (14.5) 15 ± 0.42	Pale white
26	*Aucklandia costus*	Uniseriate	Plumose	1.2 (1.3) 1.5 ± 0.15	Brownish
27	*Himalaiella heteromalla*	Uniseriate	Plumose	12 (13) 14 ± 0.55	White
28	*H. chitralica*	Uniseriate	Plumose	11.5 (12) 12.2 ± 0.67	White
29	*H. afghana*	Uniseriate	Plumose	12 (13) 14 ± 0.65	White
30	*H. albescens*	Uniseriate	Plumose	8 (10) 12 ± 0.99	Creamy–white
31	*H. chenopodifolia*	Uniseriate	Plumose	10 (11) 12 ± 0.85	White
32	*H. diffusa*	Uniseriate	Plumose	10 (11) 12 ± 0.94	White

The data in the length column indicated minimum value (mean value), maximum value, and ± standard deviation. All measurements are in millimeters (mm).

**Table 3 plants-13-03367-t003:** Carpopodium characteristics of the genus *Saussurea* and its allied genera.

S. No.	Name of Taxa	Carpopodium			
		Shape	Position	Diameter of Carpopodium (µm)	Diameter of Foramen of Carpopodium (µm)
1	*Saussurea obvallata*	Oblong	Sub-basal	255 (280) 310 ± 16.5	140 (162) 179 ± 10.05
2	*S. candolleana*	Pentagonal	Sub-basal	595 (607) 620 ± 6.89	330 (342) 355 ± 8.60
3	*S. falconeri*	Ovate	Sub-basal	900 (923) 940 ± 11.45	497 (515) 528 ± 12.49
4	*S. taraxacifolia*	Irregular	Sub-basal	410 (424) 432 ± 7.19	320 (333) 342 ± 7.81
5	*S. devendrae*	Oblong	Basal	480 (500) 512 ± 12.15	230 (245) 260 ± 13.15
6	*S. andryaloides*	Poorly developed	Poorly developed	Poorly developed	Poorly developed
7	*S. atkinsonii*	Absent	Absent	Absent	Absent
8	*S. chondrilloides*	Poorly developed	Poorly developed	Poorly developed	Poorly developed
9	*S. leptophylla*	Elliptic	Basal	262 (284) 302± 14.16	230 (242) 261 ± 11.59
10	*S. simpsoniana*	Elliptic	Basal	450 (464) 481 ± 10.22	215 (232) 250 ± 10.45
11	*S. glacialis*	Oblong	Basal	381 (400) 415 ± 10.56	263 (275) 293 ± 10.41
12	*S. gnaphalodes*	Elliptic, slightly curved	Basal	370 (386) 395 ± 7.48	302 (314) 325 ± 6.28
13	*S. bracteate*	Linear	Basal	481 (502) 521 ± 11.38	315 (326) 333 ± 5.93
14	*S. schultzii*	Irregular	Sub-basal	169 (174) 183 ± 4.89	81 (90) 93 ± 5.02
15	*S. elliptica*	Linear	Basal	815 (837) 848 ± 10.11	451 (465) 479 ± 4.09
16	*S. roylei*	Circular	Basal	561 (579.75) 596 ± 9.87	370 (383.78) 391 ± 7.59
17	*S. schlagintweitii*	Elliptic	Basal	505 (519) 529 ± 7.05	375 (380.6) 391 ± 8.98
18	*Lipschitziella ceratocarpa* var. *ceratocarpa*	Poorly developed	Poorly developed	Poorly developed	Poorly developed
19	*L. ceratocarpa* var. *astorii*	Circular	Sub-basal	265 (271) 289 ± 10.09	133 (145) 158 ± 8.56
20	*L. congesta* var. *pinnatisecta*	Absent	Absent	Absent	Absent
21	*L. congesta* var. *congesta*	Circular	Sub-basal	355 (362) 371 ± 5.09	281 (290) 296 ± 3.99
22	*Dolomiaea megacephala*	Irregular	Basal	1645 (1688) 1701 ± 17.81	928 (938) 946 ± 5.77
23	*D. macrocephala*	Absent	Absent	Absent	Absent
24	*Shangwua jacea*	Circular	Basal	450 (464) 478 ± 11.08	345 (357) 371 ± 10.59
25	*Frolovia gilesii*	Oblong	Basal	489 (500) 515 ± 12.18	333 (341) 349 ± 4.18
26	*Aucklandia costus*	Circular	Basal	584 (600) 618 ± 14.01	515 (529) 542 ± 11.98
27	*Himalaiella heteromalla*	Circular	Sub-basal	315 (321) 337 ± 9.45	125 (143) 157 ± 6.89
28	*H. chitralica*	Absent	Absent	Absent	Absent
29	*H. afghana*	Poorly developed	Sub-basal	Poorly developed	325 (333) 341 ± 5.45
30	*H. albescens*	Absent	Absent	Absent	Absent
31	*H. chenopodifolia*	Absent	Absent	Absent	Absent
32	*H. diffusa*	Poorly developed	Poorly developed	Poorly developed	Poorly developed

The data in the diameter of carpopodium and diameter of foramen of carpopodium columns indicated minimum value (mean value), maximum value, and ± standard deviation. All measurements are in micrometers (µm).

**Table 4 plants-13-03367-t004:** List of characters, scored for the cluster analysis for the genus *Saussurea s.l.* listed in Appendix A.

S. No.	Character Description	S. No.	Character Description
	**Cypsela**		**Color**
1	Length (mm)	49	Dark brown: Absent (0), Present (1)
2	Breath (mm)	50	Light brown: Absent (0), Present (1)
	**Shape**	51	Golden brown: Absent (0), Present (1)
3	Oblong: Absent (0), Present (1)	52	Greenish brown: Absent (0), Present (1)
4	Oblong–obconical: Absent (0), Present (1)	53	Blackish brown: Absent (0), Present (1)
5	Curved oblong: Absent (0), Present (1)	54	Sandy brown: Absent (0), Present (1)
6	Oblong with a depression: Absent (0), Present (1)	55	Brown: Absent (0), Present (1)
7	Oblong–lanceolate: Absent (0), Present (1)	56	Pale brown: Absent (0), Present (1)
8	Linear–oblong: Absent (0), Present (1)	57	Brown with black spots: Absent (0), Present (1)
9	Ovate: Absent (0), Present (1)	58	Dark brown–black: Absent (0), Present (1)
10	Broadly ovate: Absent (0), Present (1)	59	Brown with dark brown patches: Absent (0), Present (1)
11	Obconical: Absent (0), Present (1)	60	Dark grey: Absent (0), Present (1)
12	Fusiform–oblanceolate: Absent (0), Present (1)	61	Straw colored: Absent (0), Present (1)
13	Curved obconical: Absent (0), Present (1)		**Pappus**
14	Obovate: Absent (0), Present (1)		**Series**
15	Linear: Absent (0), Present (1)	62	Uniseriate: Absent (0), Present (1)
16	Oblanceolate: Absent (0), Present (1)	63	Biseriate: Absent (0), Present (1)
17	Quadrangular: Absent (0), Present (1)	64	Multiseriate: Absent (0), Present (1)
	**Cypsela Apex**		**Size**
18	Horns: Absent (0), Present (1)	65	Length (mm)
19	Longitudinal grooves: Absent (0), Present (1)		**Color**
	**Apical crown**	66	Brownish: Absent (0), Present (1)
20	Toothed: Absent (0), Present (1)	67	Pale brown: Absent (0), Present (1)
21	Roughly denticulate: Absent (0), Present (1)	68	Golden: Absent (0), Present (1)
22	Denticulate–lacerate: Absent (0), Present (1)	69	Golden brown: Absent (0), Present (1)
23	Mucronate: Absent (0), Present (1)	70	Golden white: Absent (0), Present (1)
24	Sinuate: Absent (0), Present (1)	71	White: Absent (0), Present (1)
	**Surface**	72	Dirty white: Absent (0), Present (1)
25	Hairs: Absent (0), Present (1)	73	Creamy white: Absent (0), Present (1)
26	Ribs: Absent (0), Present (1)	74	Brown–purplish: Absent (0), Present (1)
27	Stomata: Absent (0), Present (1)	75	Dirty white–brownish: Absent (0), Present (1)
28	Scalariform: Absent (0), Present (1)	76	Pale white: Absent (0), Present (1)
29	Appressed scalariform: Absent (0), Present (1)		**Carpopodium**
30	Faint scalariform: Absent (0), Present (1)	77	Poorly developed: Absent (0), Present (1)
31	Lineate: Absent (0), Present (1)	78	Developed: Absent (0), Present (1)
32	Irregularly lineate: Absent (0), Present (1)		**Position**
33	Lineolate: Absent (0), Present (1)	79	Basal: Absent (0), Present (1)
34	Irregularly lineolate: Absent (0), Present (1)	80	Sub-basal: Absent (0), Present (1)
35	Appressedly lineate: Absent (0), Present (1)		**Shape**
36	Sulcate: Absent (0), Present (1)	81	Oblong: Absent (0), Present (1)
37	Irregularly sulcate: Absent (0), Present (1)	82	Pentagonal: Absent (0), Present (1)
38	Reticulately sulcate: Absent (0), Present (1)	83	Ovate: Absent (0), Present (1)
39	Rugose: Absent (0), Present (1)	84	Irregular: Absent (0), Present (1)
40	Appressed colliculate: Absent (0), Present (1)	85	Elliptic: Absent (0), Present (1)
41	Colliculate in scalariform manner: Absent (0), Present (1)	86	Slightly curved: Absent (0), Present (1)
42	Colliculate in reticulate manner: Absent (0), Present (1)	87	Linear: Absent (0), Present (1)
43	Favulariate: Absent (0), Present (1)	88	Circular: Absent (0), Present (1)
44	Falsifoveate: Absent (0), Present (1)		
45	Reticulate: Absent (0), Present (1)		
46	Appressed reticulate: Absent (0), Present (1)		
47	Porate: Absent (0), Present (1)		
48	Ruminate: Absent (0), Present (1)		

## Data Availability

Data used in this study can be requested from the corresponding author via email: mmunir@kfu.edu.sa.

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
