# Peer review of "Cypsela and Pappus Morphology and Their Significance for the Taxonomic Delimitation of the Genus Saussurea DC. s.str. and Its Allied Genera (Asteraceae)"

_plants, 2024, doi:10.3390/plants13233367_

Round 1

Reviewer 1 Report

Comments and Suggestions for Authors

Some correction in text for complete

1. please check the title 

2. please check all keys

3. please check the botanical name in all figures

Author Response

Authors Response: Authors would like to extend their sincere gratitude to the reviewer for taking the time to thoroughly review our research paper. Your valuable feedback and insightful comments have greatly contributed to the improvement of our work. Your expertise and attention to detail are truly appreciated. Thank you for your dedication to advancing knowledge in our field.

Point 1. Please check the title

Authors Response: Title modified.

Point 2. Please check all keys

Authors Response: All keys checked and corrections incorporated.

Point 3. Please check the botanical name in all figures

Authors Response: All Botanical names have been corrected.

Authors Response: All suggestions mentioned in the PDF file are incorporated.

Reviewer 2 Report

Comments and Suggestions for Authors

The manuscript is well-organized and easy to understand. The SEM photographs are informative, and the identification keys based on cypsela morphology are helpful for identifying the taxa. However, the following points highlight significant weaknesses that reduce the quality of the manuscript. Due to these weaknesses, the manuscript may not contribute effectively to Plants journal.

  1. The figures and text are not correlated, and there are no references to the figures within the text.
  2. The identification keys and descriptions are based on the micro- and macromorphological features of the mentioned taxa. However, only SEM photographs are included in the manuscript. It is recommended to include additional photographs of the diagnostic characters referenced.
  3. The sample set for the taxa is too small, which limits its representation of the entire distribution range. Additional specimens for each taxon should be included, and the manuscript should be revised to reflect a more comprehensive sample set.
Comments on the Quality of English Language

-

Author Response

Comments and Suggestions for Authors

The manuscript is well-organized and easy to understand. The SEM photographs are informative, and the identification keys based on cypsela morphology are helpful for identifying the taxa. However, the following points highlight significant weaknesses that reduce the quality of the manuscript. Due to these weaknesses, the manuscript may not contribute effectively to Plants journal.

Authors Response: Authors would like to extend their sincere gratitude to the reviewer for taking the time to thoroughly review our research paper. Your valuable feedback and insightful comments have greatly contributed to the improvement of our work. Your expertise and attention to detail are truly appreciated. Thank you for your dedication to advancing knowledge in our field.

Authors Response: The identified weaknesses have been addressed.

Point 1. The figures and text are not correlated, and there are no references to the figures within the text.

Authors Response: The text is aligned with the figures, and they are referred to wherever necessary.

Point 2. The identification keys and descriptions are based on the micro- and macromorphological features of the mentioned taxa. However, only SEM photographs are included in the manuscript. It is recommended to include additional photographs of the diagnostic characters referenced.

Authors Response: In our research, we have examined the morphological features of the cypsela and pappus. To support this, we have already included images showing the complete cypsela, its surface patterns, and the carpopodium structure.

Point 3. The sample set for the taxa is too small, which limits its representation of the entire distribution range. Additional specimens for each taxon should be included, and the manuscript should be revised to reflect a more comprehensive sample set.

Authors Response: Our methodology section states that we typically analyzed 10 cypselas from each of 10 specimens per species. However, for some species, we examined fewer specimens due to limited availability. We have expanded our sample size by studying additional specimens beyond those initially reported.

Comments on the Quality of English Language       

(x) Minor editing of English language required.

Authors Response: One of the senior professors in the English department at the University of Karachi, Pakistan, thoroughly checked the English language and removed minor mistakes. Furthermore, Prof. Muhammad Munir, a co-author who is British-born and has 22 years of experience working as an advisor in a reputable agriculture-based company in Britain, has contributed to the writing of this article.

Reviewer 3 Report

Comments and Suggestions for Authors

Dear authors,

This article concerns “Cypsela morphology and its significance for the taxonomic delimitation of the genus Saussurea DC. (s. str.) and its allied genera, by Rubina Abid, Muhammad Munir, Sana Riaz, and Muhammad Qaiser. It appears apparently already in “Authorea” june 2023 https://doi.org/10.22541/ au.168569933.34247657/v1. As interesting and quite complete data I recommend it for an international audience in this journal, however several points have to be considered by the authors and a major revision is requested.

Please notice that in order to bring a broad audience to this article and to this journal, for specialists and non-specialists, the 6 major points of my comments (at the beginning) are very important (mandatory…) for a suitable value of the article. Minor points are also enhanced at the end of this review.

Sincerely yours,

The 6 major points are:

1-1     Figure 8 is of great interest, however a very increased value would be to provide the order of importance (hierarchy) of the characters used (from the most to the least important ones). I do not know if this dendrogram software provides this but you could use then another one like factorial component analysis (FCA) or hierarchical

component analysis (HCA), for instance principal component analysis (PCA) does not accept 0-1 values. Moreover, for this dendrogram colors would be much more attractive and readable; use also full names for the genera instead of abbrevations.

2-2     In the discussion, I suppose that the paragraph "Group I was further divided..." relates with your figure 8? All these cypsela features bifurcartions (divisions) should be indicated directly on the dendrogram, or if it means too many words on the same figure, you could try to use these features and divisions providing a general key of identification of all these taxa in another figure, extremely useful for all botanists. This part of the discussion is very important but it is very (too…) hard to check rapidly the key features for each taxon in the present state of your manuscript.

3-3     For all results, put all values and measurements in a table, it will be much more clear to read. In all results too, indicate the number of measurements for the values provided (normally 30 is the minimum required for statistical evaluations). Indicate also at least the standard deviation as there is certainly a variability as for all plants features.

4-4     In the discussion, detail and argument and sustain much more all sentences from "Our findings were also in accordance…".

5-     References already taken in account by the authors are of interest, however checking briefly in web of science WOS and scilit (from mdpi) with some key-words of this manuscript, other references appear, and they should be updated and used (if relevant…) in order to sustain much more and provide a larger view of these researches. Among these are the followings:

[1-7]

1.         Li, Z.; DU, G. Ecological Adaptation of the Seed Microsculptures of Saussurea from Different Altitudes (Qinghai-Tibet Plateau). Polish Journal of Ecology 2015, 63.

2.         Yuan, Q.; Bi, Y.; Chen, Y. Diplazoptilon (Asteraceae) is merged with Saussurea based on evidence from morphology and molecular systematics. Phytotaxa 2015, 236.

3.         Gavrilović, M.; Jacas, N.G.; Susanna, A.; Marin, P.D.; Janaćković, P. How does micromorphology reflect taxonomy within the Xeranthemum group (Cardueae-Asteraceae)? Flora 2019, 252.

4.         Xu, L.-S.; Herrando-Moraira, S.; Susanna, A.; Galbany-Casals, M.; Chen, Y.-S. Phylogeny, origin and dispersal of Saussurea (Asteraceae) based on chloroplast genome data. Molecular Phylogenetics and Evolution 2019, 141.

5.         Zhang, X.; Deng, T.; Moore, M.J.; Ji, Y.; Lin, N.; Zhang, H.; Meng, A.; Wang, H.; Sun, Y.; Sun, H. Plastome phylogenomics of Saussurea (Asteraceae: Cardueae). BMC Plant Biology 2019, 19.

6.         Herrando‐Moraira, S.; Group, T.C.R.; Calleja, J.A.; Chen, Y.S.; Fujikawa, K.; Galbany‐Casals, M.; Garcia‐Jacas, N.; Kim, S.C.; Liu, J.Q.; López‐Alvarado, J.; et al. Generic boundaries in subtribe Saussureinae (Compositae: Cardueae): Insights from HybSeq data. Taxon 2020, 69.

7.         Zhang, X.; Landis, J.B.; Sun, Y.; Zhang, H.; Lin, N.; Kuang, T.; Huang, X.; Deng, T.; Wang, H.; Sun, H. Macroevolutionary pattern of Saussurea (Asteraceae) provides insights into the drivers of radiating diversification. Proceedings Of The Royal Society B-Biological Sciences 2021, 288.

6-6     Although I am not english native speaker, quite many mistakes appear in the grammatical sentences and a reading by english native speaker has to be done.

Minor points are:

1 In the abstract, I do not understand "molecular decisions".

2 In the results, line 79 put "identification Key of the genera" instead of "Key to the genera", moreover this should be a figure and not a text.

3 In 2.2 line 98, indicate which is the species concerned for this key; moreover this identification key should be also a figure and not a text, as in 2.3 2.4 2.5.

4 In the discussion, put “its 6 satellite genera »?

5 Apparently tables 1 and 2 contain less characters than the description text, please homogenize all these observations.

6 In the tables and supplementary data, the use of abbreviations or not for genera is not homogeneous, please clarify these points. For instance, in suppl. data, L. line 18 is not understandable

7 Although SEM details are welcome, as it is a journal devoted to plants general view photos of the whole plants are necessary, with scale bars.

8 The captions of all SEM figures have to be much more detailed, especially for non-specialists; moreover the scale bars are for some of them almost not visible, add some clear scale bars with its value just above the bar.

Comments on the Quality of English Language

6Although I am not english native speaker, quite many mistakes appear in the grammatical sentences and a reading by english native speaker has to be done.

Author Response

Authors Response: Authors would like to extend their sincere gratitude to the reviewer for taking the time to thoroughly review our research paper. Your valuable feedback and insightful comments have greatly contributed to the improvement of our work. Your expertise and attention to detail are truly appreciated. Thank you for your dedication to advancing knowledge in our field.

The 6 major points are:

Point 1-1     Figure 8 is of great interest, however a very increased value would be to provide the order of importance (hierarchy) of the characters used (from the most to the least important ones). I do not know if this dendrogram software provides this but you could use then another one like factorial component analysis (FCA) or hierarchical component analysis (HCA), for instance principal component analysis (PCA) does not accept 0-1 values. Moreover, for this dendrogram colors would be much more attractive and readable; use also full names for the genera instead of abbrevations.

Authors Response: The dendrogram (Figure 8) was generated using software that did not allow for coloration. For this study, we used hierarchical classification through cluster analysis, assigning equal importance to all characteristics in the numerical taxonomic approach. We opted to use complete generic names rather than abbreviations for clarity.

Point 2-2.     In the discussion, I suppose that the paragraph "Group I was further divided..." relates with your figure 8? All these cypsela features bifurcartions (divisions) should be indicated directly on the dendrogram, or if it means too many words on the same figure, you could try to use these features and divisions providing a general key of identification of all these taxa in another figure, extremely useful for all botanists. This part of the discussion is very important but it is very (too…) hard to check rapidly the key features for each taxon in the present state of your manuscript.

Authors Response: The paragraph in the discussion beginning with "Group I was further divided..." corresponds to Figure 8. We could not directly label the Cypsela features on the dendrogram itself. Instead, these characteristics are detailed in Table 4 and in Table S1 (supplementary materials). The dendrogram, as presented, is sufficiently clear and informative for taxonomists to interpret without additional annotations.

Point 3-3.     For all results, put all values and measurements in a table, it will be much more clear to read. In all results too, indicate the number of measurements for the values provided (normally 30 is the minimum required for statistical evaluations). Indicate also at least the standard deviation as there is certainly a variability as for all plants features.

Authors Response: The Materials and Methods section mentioned the study of multiple Cypselas per specimen, typically involving numerous specimens from each species. It is not feasible to include individual measurements for each Cypsela in the table. Following your recommendation, we've presented the data as a range from minimum to maximum values, accompanied by mean values. We have also calculated and included standard deviations to provide a comprehensive statistical overview of the variations observed (Tables 1-4).

Point 4-4.     In the discussion, detail and argument and sustain much more all sentences from "Our findings were also in accordance…".

Authors Response: Incorporated

Point 5-.     References already taken in account by the authors are of interest, however checking briefly in web of science WOS and scilit (from mdpi) with some key-words of this manuscript, other references appear, and they should be updated and used (if relevant…) in order to sustain much more and provide a larger view of these researches. Among these are the followings:

[1-7]

Point 1. Li, Z.; DU, G. Ecological Adaptation of the Seed Microsculptures of Saussurea from Different Altitudes (Qinghai-Tibet Plateau). Polish Journal of Ecology 2015, 63.

Authors Response: Our study does not match the above study; hence, it is not incorporated.

Point 2. Yuan, Q.; Bi, Y.; Chen, Y. Diplazoptilon (Asteraceae) is merged with Saussurea based on evidence from morphology and molecular systematics. Phytotaxa 2015, 236.

Authors Response: Incorporated.

Point 3. Gavrilović, M.; Jacas, N.G.; Susanna, A.; Marin, P.D.; Janaćković, P. How does micromorphology reflect taxonomy within the Xeranthemum group (Cardueae-Asteraceae)? Flora 2019, 252.

Authors Response: Incorporated.

Point 4. Xu, L.-S.; Herrando-Moraira, S.; Susanna, A.; Galbany-Casals, M.; Chen, Y.-S. Phylogeny, origin and dispersal of Saussurea (Asteraceae) based on chloroplast genome data. Molecular Phylogenetics and Evolution 2019, 141.

Authors Response: Our study does not match the above study; hence, it is not incorporated.

Point 5. Zhang, X.; Deng, T.; Moore, M.J.; Ji, Y.; Lin, N.; Zhang, H.; Meng, A.; Wang, H.; Sun, Y.; Sun, H. Plastome phylogenomics of Saussurea (Asteraceae: Cardueae). BMC Plant Biology 2019, 19.

Authors Response: Our study does not match the above study; hence, it is not incorporated.

Point 6. Herrando‐Moraira, S.; Group, T.C.R.; Calleja, J.A.; Chen, Y.S.; Fujikawa, K.; Galbany‐Casals, M.; Garcia‐Jacas, N.; Kim, S.C.; Liu, J.Q.; López‐Alvarado, J.; et al. Generic boundaries in subtribe Saussureinae (Compositae: Cardueae): Insights from HybSeq data. Taxon 2020, 69.

Authors Response: Our study does not match the above study; hence, it is not incorporated.

Point 7. Zhang, X.; Landis, J.B.; Sun, Y.; Zhang, H.; Lin, N.; Kuang, T.; Huang, X.; Deng, T.; Wang, H.; Sun, H. Macroevolutionary pattern of Saussurea (Asteraceae) provides insights into the drivers of radiating diversification. Proceedings Of The Royal Society B-Biological Sciences 2021, 288.

Authors Response: Our study does not match the above study; hence, it is not incorporated.

Point 6-6.     Although I am not english native speaker, quite many mistakes appear in the grammatical sentences and a reading by english native speaker has to be done.

One of the senior professors in the English department at the University of Karachi, Pakistan, thoroughly checked the English language and removed minor mistakes. Furthermore, Prof. Muhammad Munir, a co-author who is British-born and has 22 years of experience working as an advisor in a reputable agriculture-based company in Britain, has contributed to the writing of this article.

We are confident enough about grammatical sentences, although some sentences have been rephrased.

Minor points are:

Point 1. In the abstract, I do not understand "molecular decisions".

Authors Response: The words molecular decisions are replaced with molecular evidences.

Point 2. In the results, line 79 put "identification Key of the genera" instead of "Key to the genera", moreover this should be a figure and not a text.

Authors Response: In taxonomic studies, keys are traditionally integrated into the main text. We have replaced the key to the genera and species with a key to the genera based on Cypsela and Pappus features, or a key to the species based on Cypsela and Pappus features. It was also suggested by Reviewer 1.

Point 3. In 2.2 line 98, indicate which is the species concerned for this key; moreover this identification key should be also a figure and not a text, as in 2.3 2.4 2.5.

Authors Response: Corrections incorporated in text.

Point 4. In the discussion, put “its 6 satellite genera »?

Authors Response: “its 6 satellite genera” is replaced with “its 6 allied genera”.

Point 5. Apparently tables 1 and 2 contain less characters than the description text, please homogenize all these observations.

Authors Response: Tables 1, 2, 3, and S1 (supplementary data) showed the characters of Cypsela, Pappus, and Carpopodium. All these observations are updated and homogenized.

Point 6. In the tables and supplementary data, the use of abbreviations or not for genera is not homogeneous, please clarify these points. For instance, in suppl. data, L. line 18 is not understandable

Authors Response: Table S1 (supplementary data) is updated and homogenized with Tables 1, 2, and 3.

Point 7. Although SEM details are welcome, as it is a journal devoted to plants general view photos of the whole plants are necessary, with scale bars.

Authors Response: Our research focuses on the macro- and micromorphology of Cypsela, rather than overall plant morphology. Thus, there is no requirement for a complete plant photograph, as Cypsela constitutes just one component of the plant.

Point 8. The captions of all SEM figures have to be much more detailed, especially for non-specialists; moreover the scale bars are for some of them almost not visible, add some clear scale bars with its value just above the bar.

Authors Response: All scanning electron micrographs include scale bars. In cases where the scale values might not be clearly visible in the image itself, we have included these measurements in the figure captions, ensuring that the scale information is readily available for all micrographs.

Comments on the Quality of English Language

Point 6. Although I am not english native speaker, quite many mistakes appear in the grammatical sentences and a reading by english native speaker has to be done.

Authors Response: One of the senior professors in the English department at the University of Karachi, Pakistan, thoroughly checked the English language and removed minor mistakes. Furthermore, Prof. Muhammad Munir, a co-author who is British-born and has 22 years of experience working as an advisor in a reputable agriculture-based company in Britain, has contributed to the writing of this article.

We are confident enough about grammatical sentences, although some sentences have been rephrased.

Round 2

Reviewer 2 Report

Comments and Suggestions for Authors

Based on the improvements made by the authors, the manuscript will be published.

Author Response

Reviewer 2

Review Report Form

Open Review

( ) I would not like to sign my review report

(x) I would like to sign my review report

Quality of English Language

( ) I am not qualified to assess the quality of English in this paper.

( ) The English is very difficult to understand/incomprehensible.

( ) Extensive editing of English language required.

( ) Moderate editing of English language required.

( ) Minor editing of English language required.

(x) English language fine. No issues detected.

Yes

Can be improved

Must be improved

Not applicable

Does the introduction provide sufficient background and include all relevant references?

(x)

( )

( )

( )

Is the research design appropriate?

(x)

( )

( )

( )

Are the methods adequately described?

(x)

( )

( )

( )

Are the results clearly presented?

(x)

( )

( )

( )

Are the conclusions supported by the results?

(x)

( )

( )

( )

Comments and Suggestions for Authors

Based on the improvements made by the authors, the manuscript will be published.

Authors response:

Thank you for your thoughtful review and valuable feedback on our manuscript. We are pleased to hear that the improvements we implemented have addressed your concerns. Your insights greatly contributed to enhancing the quality of our work. We appreciate your support and look forward to seeing our research published.

Reviewer 3 Report

Comments and Suggestions for Authors

Dear authors, 

I read this new version, however as (mainly) the two first following points are not answered and/or argumented sufficiently, I regret but this article is definitely not receivable.

The major point concerns point 3-3: (“In all results too, indicate the number of measurements for the values provided (normally 30 is the minimum required for statistical evaluations)”. “multiple” ( you use “usually “in your text, “or based on the availability of material”) have no scientific meaning at all, this number allows to understand the quality and relevance of the mean and more especially of the standard deviation. Moreover for instance “3.2(3.5)3.8±0.21 » is not understandable, it needs an explanation in the captions of each table 1-2-3.

For point 1-1, I do not mean at all the original matrix, where characters are of course treated with “equal importance to all characteristics”. I mean the result of the dendrogram made with the relative inertia (or order of importance or whatever it is called) of each character providing a hierarchy between the characters (from the most to the least important) and providing this dendrogram. If there is no relative inertia there is no dendrogram. This relative inertia is extremely useful in all kinds of classifications. The identification key that you provide (very useful) relates with the qualitative appreciation of your material while dendrogram (very useful too) relates with quantitative mathematical approach (thus more objective?). Both approaches (their results may be (very) different…) are of course of course receivable in taxonomy but this double approach has to be much more detailed and discussed than in the present text. The best (!) solution (at least the most objective) is usually to provide a “synthetic” key using both approaches.

For point 4-4, it is still very weakly developed.

For figure 8, the abbreviations of genera names are still totally un-understandable as not homogeneous at all, it has to be read and understood rapidly.

For point 2-2, still I cannot see Figure 8 appearing in the discussion. Moreover “The dendrogram, as presented…” as to be understood rapidly by a reader, which is not at all the present case.

For my minor point 2, although “In taxonomic studies, keys are traditionally integrated into the main text”, they are also frequently put in figures, which makes the text much more understandable.

For my minor point 5, I cannot see clearly all your corrections in yellow color

I regret the absence of photos of the whole plants, making this article much more attractive especially for non-specialists

Comments on the Quality of English Language

Although the english is quite fluent, I still think that more extensive english corrections have to be done 

Author Response

Reviewer 3

Review Report Form

Quality of English Language

( ) I am not qualified to assess the quality of English in this paper.
( ) The English is very difficult to understand/incomprehensible.
( ) Extensive editing of English language required.
( ) Moderate editing of English language required.
(x) Minor editing of English language required.
( ) English language fine. No issues detected.

Yes

Can be improved

Must be improved

Not applicable

Does the introduction provide sufficient background and include all relevant references?

(x)

( )

( )

( )

Is the research design appropriate?

( )

( )

(x)

( )

Are the methods adequately described?

( )

( )

(x)

( )

Are the results clearly presented?

( )

( )

(x)

( )

Are the conclusions supported by the results?

( )

(x)

( )

( )

Authors would like to extend their sincere gratitude to the reviewer for taking the time to thoroughly review our research paper. Your valuable feedback and insightful comments have greatly contributed to the improvement of our work. Your expertise and attention to detail are truly appreciated. Thank you for your dedication to advancing knowledge in our field.

Comments and Suggestions for Authors

Dear authors, 

I read this new version, however as (mainly) the two first following points are not answered and/or argumented sufficiently, I regret but this article is definitely not receivable.

The major point concerns point 3-3: (“In all results too, indicate the number of measurements for the values provided (normally 30 is the minimum required for statistical evaluations)”. “multiple” (you use “usually “in your text, “or based on the availability of material”) have no scientific meaning at all, this number allows to understand the quality and relevance of the mean and more especially of the standard deviation. Moreover for instance “3.2(3.5)3.8±0.21 » is not understandable, it needs an explanation in the captions of each table 1-2-3.

Authors Response:

  • The specimen size for measurements are included in line No. 313 and 314.
  • Point regarding “3.2(3.5)3.8±0.21”, is now explained in the footnotes of Tables 1, 2, and 3.

For point 1-1, I do not mean at all the original matrix, where characters are of course treated with “equal importance to all characteristics”. I mean the result of the dendrogram made with the relative inertia (or order of importance or whatever it is called) of each character providing a hierarchy between the characters (from the most to the least important) and providing this dendrogram. If there is no relative inertia there is no dendrogram. This relative inertia is extremely useful in all kinds of classifications. The identification key that you provide (very useful) relates with the qualitative appreciation of your material while dendrogram (very useful too) relates with quantitative mathematical approach (thus more objective?). Both approaches (their results may be (very) different…) are of course of course receivable in taxonomy but this double approach has to be much more detailed and discussed than in the present text. The best (!) solution (at least the most objective) is usually to provide a “synthetic” key using both approaches.

Authors Response: The synthetic key is provided in Figure 9 (page 13).

For point 4-4, it is still very weakly developed.

Authors Response: We have modified the discussion in the light of reviewer’s suggestions.

For figure 8, the abbreviations of genera names are still totally un-understandable as not homogeneous at all, it has to be read and understood rapidly.

Authors Response: Figure 8 is modified as suggested.

For point 2-2, still I cannot see Figure 8 appearing in the discussion. Moreover “The dendrogram, as presented…” as to be understood rapidly by a reader, which is not at all the present case.

Authors Response: Figure 8 in mentioned in the text (line No. 198).

For my minor point 2, although “In taxonomic studies, keys are traditionally integrated into the main text”, they are also frequently put in figures, which makes the text much more understandable.

Authors Response: We have included the synthetic key. Please see Figure 9 (page 13).

For my minor point 5, I cannot see clearly all your corrections in yellow color.

Authors Response: We have incorporated the most relevant below two references in the manuscript:

Yuan, Q.; Bi, Y.; Chen, Y. Diplazoptilon (Asteraceae) is merged with Saussurea based on evidence from morphology and molecular systematics. Phytotaxa 2015, 236.

Gavrilović, M.; Jacas, N.G.; Susanna, A.; Marin, P.D.; Janaćković, P. How does micromorphology reflect taxonomy within the Xeranthemum group (Cardueae-Asteraceae)? Flora 2019, 252.

I regret the absence of photos of the whole plants, making this article much more attractive especially for non-specialists.

Authors Response: Plant photographs are added. Please see Figure 10 (page 14).

Comments on the Quality of English Language

Although the english is quite fluent, I still think that more extensive english corrections have to be done.

Authors Response: The English language is thoroughly check by:

  1. Muhammad Adnan Shahid, Assistant Professor, Horticultural Science Department, North Florida Research and Education Center, University of Florida/IFAS, Quincy, FL 32351, USA. Email: mshahid@ufl.edu
  2. Muhammad Iftikhar Hussain, Assistant Professor, Department of Plant Biology and Soil Science, Universidade de Vigo, 36310-Vigo, Spain. Email: mih786@gmail.com
